# A survey of UK beekeeper's *Varroa* treatment habits

**Alexandra Valentine**¤, **Stephen J. Martin** *

School of Science, Engineering and Environment, The University of Salford, Manchester, United Kingdom

¤ Current address: School of Natural Sciences, National University of Ireland Galway, Galway, Ireland
* s.j.martin@salford.ac.uk

## Abstract

The global spread of the parasitic mite *Varroa destructor* instigated a substantial decline in both managed and feral honeybee (*Apis mellifera)* colonies mainly across the Northern hemisphere. In response, many beekeepers began to treat their colonies with chemical acaricides to control mite populations in managed colonies. However, some countries or beekeepers allowed their bees to develop mite-resistance by adopting a "treatment-free" approach, rather than using selective breeding programs. Yet, the distribution and proportion of beekeepers either treating or not within the United Kingdom (UK) is unknown, as it is in most Northern hemisphere countries. Therefore, the aim of this study was to conduct a beekeeper survey to determine the current treatment strategies within the UK. We gathered 2,872 beekeeper responses from an estimated 30,000 UK beekeepers belonging to 242 bee-associations in the winter of 2020/21. The survey indicated that the majority (72–79%) of UK beekeepers are still treating their bees for *Varroa*, typically twice-yearly using chemical-based methods. Six percent or 1,800 UK beekeepers were treatment-free for six years or more. This is reflected by our finding that 78 associations out of 242 consist of responders who entirely treated, while only four associations had more than 75% of their members that were non-treating. Overall treatment status was not affected by association currently. Using the baseline data from this survey it will be possible in the future to observer if a shift towards treatment-free beekeeping occurs or not.

## Introduction

Honeybees are key pollinators of crops and wildflowers [1]. They contribute globally more than €53 billion annually to the economy [2]. Still, honeybees are presently facing multiple stressors such as habitat destruction, pests, and pathogens. Across the Northern hemisphere the spread of the ecto-parasitic mite *Varroa destructor* (*Varroa*) and the key viral pathogen it transmits (Deformed Wing Virus) is the number one threat faced by beekeepers over past decades, causing an increase in over-wintering losses [3,4]. In the UK, this was evidenced in the very high losses of managed colonies and the near eradication of feral colonies within a few years of *Varroa* becoming established [5]. Over-wintering colony losses have been declining in the UK during the past decades (S1 Fig) since *Varroa*-treatments became widespread [6].

**Data Availability Statement:** All relevant data are within the manuscript and its Supporting information files.

**Funding:** The authors received no specific funding for his work.

**Competing interests:** The authors have declared that no competing interests exist.

Although, colony losses remain consistently high in the USA [7], which is unexpected due to the high levels of treatment carried out by beekeepers to control *Varroa* populations in both countries. Typically, a variety of acaricides are primarily used to control *Varroa* populations, but with the development of acaricide resistance [8] alternative commercially based treatments established around thymol and organic acids (oxalic and formic) have become increasingly popular. More time consuming and often less effective biomechanical treatment methods can also be used, such as powdered sugar and drone trapping. The application of Integrated Pest Management (IPM) techniques is suggested to be the most effective way [9] to treat mites since it involves a variety of methods applied in rotation throughout the year determined by mite-infestation levels through routine observations and mite monitoring carried out by beekeepers.

In contrast, some countries like South Africa [10] and Cuba [11] took a decision not to treat *Varroa* and allowed mite-resistance to appear naturally. In both countries 1,000's of colonies were lost initially, but losses declined after several years as resistance to the mite developed. Whereas, in Brazil the evolution of mite-resistance by Africanised bees (*Apis mellifera scutellata*) was not observed and any losses were probably masked by the losses from untreated non-resistant European honeybee colonies, although there is no published evidence to support or contradict this idea. Despite this *Varroa* resistance rapidly became widespread in these countries, hence treatment-free beekeeping has already been established for several decades.

More recently in mainland Europe [12,13], the UK [14], and USA [15] it appears an increasing number of mite-resistant populations are being managed treatment-free. Allowing honeybees to develop natural resistance will foster a long-term solution to the *Varroa* problem.

Resistance is defined as any situation in which *Varroa* populations are maintained at a suitable level for the long-term survival of the honeybee colonies [16]. Recently, three key traits (cell recapping, mite infertility and brood removal) have been associated with natural resistance in almost all resistant populations studied in different countries [17]. In the UK the government advice is to treat *Varroa* with either biotechnical methods, with registered varroacides or a combination of them [18]. Despite this, anecdotal evidence from beekeepers suggests that numbers of non-treating beekeepers are more than expected [19].

The aim of this study was therefore to conduct an online survey of UK beekeepers via their associations to assess individual treatment habits, as it was the most efficient way to reach beekeepers easily. The survey will provide crucial empirical data to support or refute common perceptions around treatment and non-treatment beekeeping practices.

## Materials and methods

### The survey

The survey was constructed using Google Forms as it allowed an unlimited number of responses whilst also allowing the incorporation of the University logo to add credibility to the survey. The survey consisted of a brief description outlining the study, its aims, and our definition of treatment, which was, "any form of external or additional control administered to bees by beekeepers aimed at reducing *Varroa* numbers". Then followed six questions: 1) association name, 2) number of colonies, 3) if they treat or not, 4) number of times a year they treat, 5) number of years since last treatment, 6) type of treatment (see S2 Fig for survey). Answers were either multiple-choice questions or open questions that all helped assess the beekeeper's treatment habits. The survey was kept short in duration to increase the response rates [20].

The contact details for 325 beekeeping area associations across the UK (Fig 1A) were obtained via the four UK Beekeeper Association websites (British, Scottish, Welsh and Northern Ireland). Initial contact was made by email with 303 of them; outlining the study aims to

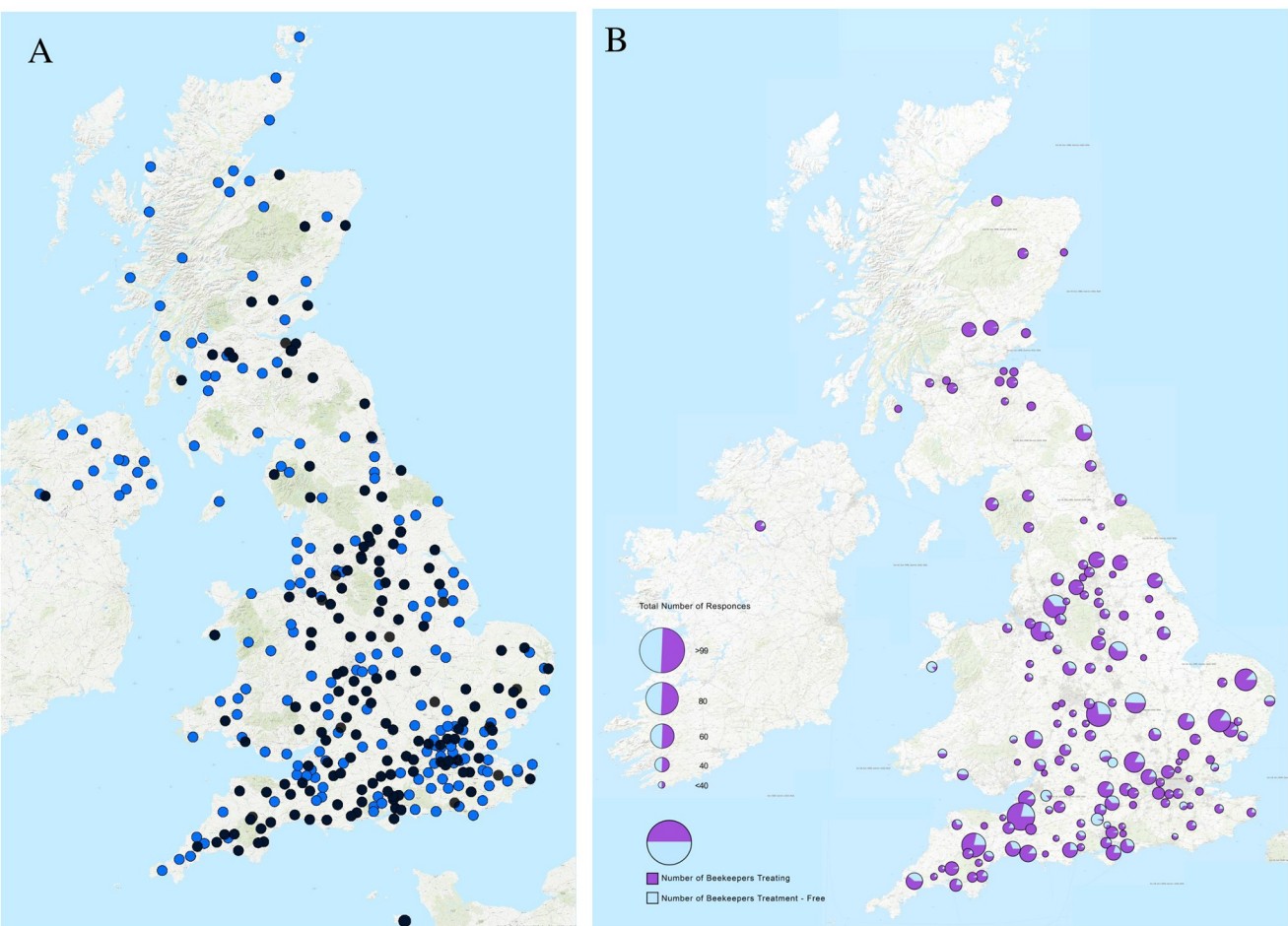

**Fig 1.** A) Distribution of all 325 UK beekeeping area associations (blue dots) and the 158 area associations with five or more respondents that we used in the spatial study (black dots). B) Distribution and relative proportion of *Varroa*-treating (purple) and treatment-free beekeepers (pale blue) at an associations level across the UK. Larger pies indicate more responses from that association. Map image is the intellectual property of Esri and is used herein under license. Copyright © 2020 Esri and its licensors. All rights reserved. Esri, USGS | Esri UK, Esri, HERE, Garmin, FAO, NOAA, USGS.

determine general interest. The responding associations were then sent the survey online link and asked to forward it to their respective members. Beekeepers were given ten weeks from 23rd Dec 2020 to complete the survey. A small number of beekeepers did not complete every section of the survey fully, so for each analysis the precise number of responses obtained from that part are given. Amateur bee-associations were chosen as they contain around 30,000 members in the UK.

## The survey analysis

After the survey closed, all 2,897 responses from 243 associations were exported into an excel spreadsheet. The 25 responses from the Isle of Man, which remains a *Varroa*-free island were excluded along with the 16 beekeepers with zero colonies leaving 2,856 for further analysis. A total of 67 beekeepers did not either belong to or failed to put down an association. As over 20 types of *Varroa* treatment were reported, we grouped them into single methods (Chemical, Biomechanical and Natural) and mixed methods. To estimate the number of colonies surveyed we multiply the median colony group size, or 30 for the 30+ group (Table 1) by the number of beekeepers in that group.

**Table 1. The number of colonies, shown in four groups, managed by the responding beekeepers.** The percentage of each group relative to the total is also given. Also the number of treatment-free beekeepers with that group are presented along with the percentage of the that group.

| Colony number groups | No. of beekeepers (% of respondents) | No. of treatment-free beekeepers (% of treatment-free beekeepers in group) |
|---|---|---|
| 1–5 | 1,912 (67%) | 430 (23%) |
| 6–15 | 716 (25%) | 121 (17%) |
| 16–30 | 139 (5%) | 28 (20%) |
| 30+ | 89 (3%) | 17 (19%) |
| Total | 2,856 | 596 (21%) |

## Spatial analysis

While conducting previous research into *Varroa*-resistance colonies [14] it appeared that they existed in many parts of the UK. To investigate this maps were created using ArcGIS Pro (v. 10.8.1) software, showing the location of all UK area associations and those that participated (Fig 1A). To investigate if any spatial patterns in treatment-free vs treated existed, the proportion of beekeepers falling into either category were calculated from each association that had at least five responses to avoid illegible pie diagrams on the figure (Fig 1B).

## Statistical analysis

A GLM was conducted in R [21] using treatment status as the response variable, association as the fixed effect and colony number as the random effect. A binomial family was used to fit the model due to only two variants of the response variable (i.e., treating or treatment-free).

The median group size or 30 for the 30+ group were used for colony group size in the GLM.

## Results

The 2,856 beekeepers who responded represents almost 10% of the estimated 30,000 members belonging to the four UK beekeeping associations and were widely distributed across the UK (Fig 1A). We estimated that we surveyed around 21,200 colonies that is again around 10% of the 220,000 colonies estimated by the Center for Ecology and Hydrology to be in the UK.

The majority (67%) of beekeepers that responded managed between 1 to 5 colonies and only 3% had over 30 colonies (Table 1). Our data indicated that across the five colony size groups the proportion of treatment-free beekeepers was greatest in the 1–5 group, but all other groups were within 5% of that group (Table 1). The GLM (Table 2) showed that treatment status was not affected by association but did indicate a significant effect of colony group size on treatment status. However, as the 95% CI (-19.559–15.493) contains 0 then we can conclude that there is not a strong significant relationship. Due to such high variability in the data, we

**Table 2. A generalized linear model (GLM) showing the association had no effect on colony number but there was a significant effect of colony number on treatment status.** The range of the Z and Pr values for associations are given rather than the long list of individual values.

| | Estimate | Std. error | Z value | Pr(>|z|) |
|---|---|---|---|---|
| Intercept | -1.830E+01 | 2.463E+03 | -0.007 | 0.994 |
| Colony number group | -1.815E-02 | 7.951E-03 | -2.282 | 0.0225* |
| Association | -9.647E-02-3.703E+01 | 2.463E+03–6.972E+03 | 0–0.012 | 0.993–1 |

*<0.05.

**Table 3. Number and percentage of responses for the time since beekeepers last treated their colonies for *Varroa*.** Enclosed in parentheses is the year ranges in which the last treatments were administered.

| Last treatment administered | N | Percentage of total (%) |
|---|---|---|
| <1 year (2020–2021) | 2,046 | 72 |
| 1–5 years (2016–2020) * | 536 | 19 |
| 6–10 years (2016–2011) | 108 | 4 |
| 10+ (2011 or earlier) | 65 | 2 |
| Never administered a treatment | 100 | 4 |
| Total | 2,856 | |

* Some beekeepers in this group may treat if their colonies become overwhelmed with mites, but at the time of the survey it indicates the intention is not to treat their colonies.

conclude that there is not a statistically significant relationship between colony number and treatment status.

## Treatment-free beekeeping

A total of 596 (21%) beekeepers stated they were not treating, and 2,260 (79%) beekeepers were treating their colonies against *Varroa*. When asked about the duration elapsed since their last treatment was applied, 72% had treated within the last year and 173 (6%) responders had not treated for 6 years or more (Table 3). The spatial distribution (Fig 1B) indicates Scottish beekeepers were almost all treating and only four associations had more than 75% of treatment-free beekeepers as members, otherwise treatment free beekeepers were spread throughout England. This indicates the widespread distribution of beekeepers attempting treatment free approaches particularly in England.

## Treatments used in the UK

An estimated 4,093 treatments per year were administered by 2,238 beekeepers with the aim of reducing *Varroa* numbers. The majority (70%) are treated once or twice a year (Table 4) using a single chemical method (Table 5). The most popular chemical treatment method is oxalic acid, followed by commercially produced thymol, and amitraz. The current study found 78% of beekeepers use chemical treatments (oxalic acid, thymol etc), 3% use biomechanical methods (drone brood removal, sugar dusting etc) and less than 1% use other methods (rhubarb leaves, etc.) (Table 5). Overall, 80% use a single method of treatment and only 20% use a combination of treatment methods (Table 5).

**Table 4. Treatment application data outlining the number of responses for each application category and subsequent number of treatment applications applied to *Apis mellifera* with the aim of reducing *Varroa destructor* mite numbers.**

| Applications per year | N | Percentage of total (%) |
|---|---|---|
| 0 (i.e., not treating) | 596 | 21 |
| 1 | 864 | 30 |
| 2 | 1,148 | 41 |
| 3+ | 226 | 8 |
| Total | 2,834* | |

* a small number of beekeepers failed to report what treatment type they used.

**Table 5. The number of treatment applications per year from 2,238 beekeepers across the UK grouping treatments into single compounds or methods, along with beekeepers those using a combination of methods.**

| Treatment Category | % of total treatments |
|---|---|
| **Active ingredients used in single applications 80%** | |
| **Chemical** | |
| Oxalic acid | 31.4 |
| Thymol | 19.3 |
| Amitraz | 18.3 |
| Formic acid | 10.8 |
| Tau-fluvalinate | 3.7 |
| Flumethrin | 0.2 |
| Others | 0.2 |
| **Biomechanical** | |
| Drone brood removal | 5.4 |
| Sugar/flour dusting | 5.1 |
| Shook swarm | 1.6 |
| False/Mesh floor | 1.5 |
| Bee gym | 0.6 |
| Brood breaks | 0.4 |
| Changing frames | 0.2 |
| Queen trapping | 0.2 |
| **Natural methods** | |
| Rhubarb leaves | 0.3 |
| Thyme oil | 0.2 |
| Crushed garlic | 0.1 |
| Homeopathic sprays | 0.2 |
| Scented leaves | 0.1 |
| Eucalyptus oil | 0.1 |
| Other essential oils | 0.1 |
| **Mixed treatments 20%** | |
| Chemical/ biomechanical | 17.1 |
| Biomechanical/natural | 0.3 |
| Chemical/natural | 0.4 |
| Chem/bio/natural | 0.2 |
| **Total number of treatments** | **4,093** |

## Discussion

Based on 2,856 beekeeper responses from 243 UK beekeeping areas, the proportion of bee-keepers not-treating ranged between 21–28% e.g., if the beekeeper can only choose between treatment or treatment-free (21%), or 28% of beekeepers if you include those that had not treated in the last year. Those not-treating for over 6 years represented 6% of responders. Based on this survey, that would mean 1,800 of the 30,000 estimated UK beekeepers are truly treatment-free having not treated their colonies for *Varroa* for six years or more. Around 100 of these beekeepers are in a single region in North Wales [22] with many belonging to the Lleyn and Eifionydd Beekeepers' Association. Likewise, in Swindon, a small beekeeper group have kept treatment-free colonies since 1995 [23]. In the UK there are active and growing "treatment-free" communities. For instance, the Westerham beekeepers are approaching their 5th year of becoming treatment free and starting to bring in neighboring bee clubs. Finally, the

'Natural Beekeeping Trust' was established in 2009 and now has links to 35 like-minded groups. We had 18 responses from their members all falling into the 'never-treated group', along with 16 beekeepers that failed to provide an association. These 'natural beekeepers' and those not-treating for *Varroa* have not always been welcomed by those adopting treatment regimens [24]. The 'never-treated group' represent 4% of the respondents but the length of time they have not been treating is unknow. These treatment-free beekeepers occur across all colony group sizes (Table 1).

The annual BBKA (British Beekeepers Association) overwinter survival survey found that in 2020/21, 27% and 37% of 2,950 randomly selected members from a pool of 26,407 did not treat from August to September or from October to April respectively [6]. In 2019/20 the BBKA survey found 25% and 38% of treatment-free beekeeping during Aug-Sept and Oct-Apr, respectively [25]. In both years the BBKA survey found treatment-free beekeeping was present in all English regions, as was found by this study (Fig 1B). This is important, since the two surveys used different sampling methods. The BBKA and this study surveyed similar numbers of beekeepers, except the BBKA survey targets members randomly selected from their membership each year, while this study approaches the beekeepers via their associations using email, so they are self-selecting. Despite this the two outcomes are similar in many aspects indicating that our survey has not disproportionally been returned by treatment-free beekeepers that may be more active online.

Nonetheless, the majority (72–79%) of UK beekeepers are still treating their colonies to control *Varroa* numbers. We found beekeepers in the UK are predominantly using a single or bi-annual chemical treatment regime. The order of popularity starting with the most common is oxalic acid, thymol, amitraz and formic acid. Only 20% of beekeepers in this study are adopting a combination of methods approach (Table 5).

The popular chemical *Varroa* treatments like formic and oxalic acids reported in this study and other UK surveys [6,25] are also the preferred methods used in Europe [26] and the USA [27]. These compounds have a high efficacy but without the stigma of synthetic compounds or mite resistance, which could explain why many beekeepers are choosing to adopt these methods [28]. However, the impact of these "natural" treatments should not be ignored. Whilst thymol is thought to be unproblematic at temperatures between 5°C-9°C, high mortality levels have been observed when temperatures exceed 27°C [26,29,30].

The success of any treatment method or treatment-free beekeeping is determined by their overwinter losses. Over the last 14 years in the UK over-wintering annual colony losses (S1 Fig) have ranged from over 30% to 8% with an average of 18%. This falls to 14% when the three spikes related to unusually cold and wet winters are removed (data derived from BBKA annual survey data collected by D. Aston that is based on 2,500–3,500 beekeepers each year [e.g. 6,25]). Principal causes of colony losses reported by beekeepers in 2021 were queen-related problems (24%), isolation starvation (21%), weather-related 17%, and *Varroa* was just 4% [6] reflecting the situation that the mite is no longer considered a major problem by beekeepers currently in the UK. However, *Varroa* could be contributing to some of the other colony losses indirectly. In 2019–2020 the BBKA annual survey reported both losses in the region as well as the percentage of colonies not treated in each region [25].

Beekeeper observations by the Oxford shire Natural Beekeeping Association [31] (ONBA) and the treatment-free group in Northwest Wales [22] both found their colony survival rates were on par with those from the BBKA, but independent scientific studies are required to see if *Varroa*-resistance effects the rate of colony losses.

One of the dangers of switching over to an intervention free-treatment regime involves a period of high winter losses whilst the bees develop resistance [32]. This is due to re-invasion from collapsing colonies to surviving surrounding colonies [24] or by visits from infested non-

natal bees [33]. Several strategies have been adopted at both local and national level to encourage mite-resistant honeybees. Catching free-living swarms from locations where they appear to have persisted for many years has been a successful strategy in the UK [22] and Hawaii [34]. Others have reduced the frequency of mite-treatments by selectively treating colonies with high mite levels, or to use less efficient biomechanical methods only when required.

Already many countries (South Africa, Brazil, Mexico, Cuba etc), along with beekeepers in the UK and elsewhere have been able to stop treating as their honeybees have learnt to detect mite infested cells and remove the pupa to prevent mite-reproduction, which leads to decreased mite fertility and population size. However, the majority (72–79%) of UK beekeepers are still treating and it will be many years before most beekeepers in the UK and elsewhere can stop treating for *Varroa*. In the USA a recent survey found only around 63 (3%) of 2275 respondents stated no advantages to *Varroa* management and 92% of this group did not treat for *Varroa* [35], a very different situation than found currently in the UK. Most beekeepers in the Northern hemisphere have long wished for a silver bullet for the *Varroa* problem, however, it turns out that the bullet is their own bees. Beekeepers just need to give their bees time to develop mite resistant as has been done so successfully elsewhere in the world.

## Supporting information

**S1 Fig. The annual overwintering colony losses in England and Wales recorded by the BBKA survey conducted by Dr D. Aston.** The data was compiled from the annually published data contained in the BBKA newsletter. We have added in the 5-year rolling average.
(DOCX)

**S2 Fig. A copy of the online survey as was seen by participants.** The university logo has been implemented to increase the credibility of the survey.
(DOCX)

## Acknowledgments

We thank all the UK beekeepers that submitted a survey, help from their associations and G. Webb of University of Salford for proofreading the original submission. The original data is available on dryad at https://doi.org/10.5061/dryad.xksn02vkn.

## Author Contributions

**Conceptualization:** Alexandra Valentine, Stephen J. Martin.

**Data curation:** Alexandra Valentine.

**Formal analysis:** Alexandra Valentine.

**Funding acquisition:** Stephen J. Martin.

**Investigation:** Alexandra Valentine.

**Methodology:** Alexandra Valentine, Stephen J. Martin.

**Project administration:** Alexandra Valentine, Stephen J. Martin.

**Resources:** Stephen J. Martin.

**Supervision:** Stephen J. Martin.

**Validation:** Alexandra Valentine.

**Visualization:** Alexandra Valentine.

**Writing – original draft:** Alexandra Valentine.

**Writing – review & editing:** Alexandra Valentine, Stephen J. Martin.

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
