## [Decision Letter · Decision Letter 0]

29 May 2022

PONE-D-22-05221A survey of UK Beekeeper’s Varroa treatment habitsPLOS ONE

Dear Dr. Martin,

Thank you for submitting your manuscript to PLOS ONE. After careful consideration, we feel that it has merit but does not fully meet PLOS ONE’s publication criteria as it currently stands. Therefore, we invite you to submit a revised version of the manuscript that addresses the points raised during the review process.

Dear Steve,Although all referees thought your paper and its objectives interesting, they felt that some important informations were missing with regards to the methods and analyses used and whether this would influence the results and conclusions of the paper. One main issue is how colony loss was assessed (if it was) and how numbers were extracted from the survey. Please attend to the comments and explicit your answer in a rebuttal lettercheers

We look forward to receiving your revised manuscript.

Kind regards,

Nicolas Chaline

Academic Editor

PLOS ONE

Journal Requirements:

Reviewers' comments:

Reviewer's Responses to Questions

**Comments to the Author**

1. Is the manuscript technically sound, and do the data support the conclusions?

Reviewer #1: Yes

Reviewer #2: Partly

Reviewer #3: Partly

2. Has the statistical analysis been performed appropriately and rigorously? 

Reviewer #1: Yes

Reviewer #2: I Don't Know

Reviewer #3: Yes

3. Have the authors made all data underlying the findings in their manuscript fully available?

Reviewer #1: Yes

Reviewer #2: No

Reviewer #3: Yes

4. Is the manuscript presented in an intelligible fashion and written in standard English?

Reviewer #1: Yes

Reviewer #2: No

Reviewer #3: Yes

5. Review Comments to the Author

Reviewer #1: The submitted survey of UK beekeeper´s varroa treatment habits presents interesting new data and fills a gap in our knowledge on recent beekeeping practice. The study is well presented with a clear description of the data sampling and evaluation procedure. The discussion picks up relevant links to related surveys and the establishment of mite resistance in bee populations depending on the treatment strategies. The length of the different chapters is appropriate and well balanced.

Some confusion may arise about the number of respondents for the data in table 3. The table reports on a total of 2,868 responses from 2,897 responses in general (L116). If 25 responses from the Isle of Man have been excluded, some of the 16 participants with zero colonies (L146f) seem to be included here. However, this wouldn´t make any sense for my understanding. The authors should give a precise description on this.

L102 refers to Fig S1 which is missing in the supplied manuscript.

In L120 Table S1 is mentioned but it should be Table 5 as far as I understand.

In L 264 the second “by” has to be erased.

Reviewer #2: The paper has the potential to be interesting, but is currently missing the access to the data and the supplemental information, so difficult to assess. Also the conclusions are very broad and the data presented don't support the conclusions the authors make, as the survey did not include any assessment of colony losses. Without distinguishing what the actual loss rates of the treatment free beekeepers have, it is not possible to state that the bees are actually resistant to varroa or have a high tendency for survival. The higher colony numbers actual suggest the opposite.

Also there are a number of odd sentence structures and grammatical errors. I have made note of the mores specific comments in a separate document.

Reviewer #3: The study aims to estimate the number of beekeepers in the UK who are not treating their colonies for varroa, and to estimate whether the existence of untreated and resistant colonies in the UK could be assumed. Information on beekeeping practices are currently limited in the scientific literature, and this question and the data the study provides are really relevant and valuable. The article is generally clear and well written. The methodological choices are globally sound and with a high number of respondents to the survey, which is important to note. Still some information should be added in the methodological section, and the limits induced by some methodological choices should be identified and discussed as they can have important impacts on the interpretation of the results. Also, the findings should be discussed regarding the scientific literature on the subject. For these reasons I recommend major revisions.

Major comments

Some methodological limits should be identified and discussed, especially i) the possible influence of the sampling through the beekeeping associations (are these associations involved in prescribing varroa treatments?) and ii) the lack of information that the survey provides about the colony losses and about the beginning year of the beekeeping activity. Without this last information, the share of beekeepers who have “never treated” cannot be interpreted as it is done currently regarding the possible existence of surviving colonies (those beekeepers could have started recently and their colonies may not have faced this absence of treatment for varroa for a long time). It should not be suggested that these colonies had never known any treatment (it could be the case, but the data of the survey do not allow to conclude on this question).

The existence of colonies surviving without treatment for varroa and the possible losses that the absence of treatment may induce should also be discussed regarding the scientific literature on these questions.

Other comments

L. 57: “due to almost universal treatment by beekeepers”: this statement should be either nuanced or supported by a reference about American beekeepers’ practices. E.g. see Thoms et al. (2019 - https://doi.org/10.1007/s13280-018-1130-z), which indeed reveals a high percentage of American backyard beekeepers reporting not to treat for varroa.

L. 100-105: If I understand correctly, there was no question about the colony survival (cf. L100-105). It would have been a valuable information, as the emergence of resistant colonies is only possible if the non-treated colonies survive (in the case of backyard beekeepers replacing regularly their colonies that died from varroa, these non-treated colonies would not be more resistant than other).

L. 106-113. As the beekeeping associations who forward the survey to their adherents play a major role in the sampling, it would be useful to provide more information about these associations (as such beekeeper associations and their role can differ from one association or one country to the other). Especially, can these associations interfere with their adherents’ treatment choices (e.g. by advising or not advising to treat for varroa)?

L. 123. One of the main analysis choices is the investigation for a possible spatial pattern. As many other factors could possibly influence beekeepers’ choice about their treatment for varroa (beekeeping experience, age, etc), this hypothesis of a spatial pattern could be explained and justified.

L. 130. Some biomechanicals and natural methods that were reported by beekeepers (cf. Table 5) are not really efficient against varroa. As the study focuses on the possible existence of untreat / resistant colonies and not on the choice of beekeepers to treat or not to treat (reasons for such choices, etc), these “low-efficacy” treatments could have been considered jointly with the treatment-free group, or a third variant for the response variable could have been considered. So the choice to gather all the treatments together in a single response variable could be explained.

L. 133. As one of the questions of the study is the existence of untreated and possibly resistant colonies, it would be useful to know how many colonies in total these 2,872 beekeepers manage and what percentage of the total number of colonies in the UK it represents.

L. 141-144. This passage is not very clear, especially what is supposed to “explained why there was significant increase in treatment-free beekeepers as the numbers of colonies they managed increased”. It would be useful to rephrase or to complete it.

L. 146 / Table 1. The third column heading is not very clear: does it represent the percentage of colonies of the group (1-5,…) which are not treated? So 73% of the colonies of the 16-30 group are not treated?

An additional column providing the percentage of treatment-free beekeepers in each of the groups would be welcome. Besides, the final percentage of untreated colonies (gathering all groups) should be added.

L. 160-162. Grouping the beekeepers who indicated that they have "never treated" with those indicating a specific number of years since the last treatment (here 10 years and more, or 6 years and more L. 201) is questionable as the date of installation of the beekeepers was not in the survey and is not known. Some of them could have started beekeeping recently, and the fact that they have “never treated” for varroa does not presume that their colonies had to face a significant period without treatment. This group of beekeepers with an unknown number of years without treatment should be considered separately and the confusion with the groups where a long treatment-free period is known should be avoided.

L. 163-164. The fact than some association gather more than 75% of treatment-free beekeepers raises questions about the role of these associations, and about the possible exchanges related to varroa treatments that its members may have. Even if it was not the objective of the study to understand the determinants of the absence of treatment, it would be interesting to discuss the possible role of associations on this point given their central place in the survey sample.

L. 178. “the majority are treated”: the majority of colonies?

L. 180-184. It would be useful to add the global percentage of beekeepers not using any treatment and to more clearly distinguish if the percentages given for the types of treatments are exclusive of each other or not. E.g. can the 3% of beekeepers using biomechanical methods be also in the 78% using chemical? Or are they only using biomechanicals methods?

L. 201. See comment on L. 160-162

L. 202. The extrapolation of the study results to the 30,000 estimated UK beekeepers should be supported by a discussion about the representativity of the sample. That joins the question of giving information about the beekeeping associations (see comment on L 103-115), and about their representativeness (are they some apicultural practices which may differ between members of these associations and beekeepers who are not members, e.g. if the associations are involved in prescribing treatments for varroa? Are the members of these associations representing a large share of the UK beekeepers?)

L. 257-260. As varroa is usually considered in scientific and technical literature as an important factor of colony losses, some scientific references about the place of varroa in colony losses need to be added and the technical references which are the only provided references should be discussed regarding these scientific references.

L. 267. It would be useful to precise “in Hawaii” and not only “in the USA” as the context of an island can be specific.

Fi. 1. A. A legend with the number of respondents corresponding to the different pie sizes is needed here.

6. PLOS authors have the option to publish the peer review history of their article (what does this mean?). If published, this will include your full peer review and any attached files.

Reviewer #1: No

Reviewer #2: No

Reviewer #3: No

---

## [Author Response · Author response to Decision Letter 0]

1 Jul 2022

Data is said to be publicly available, but no link included. Same goes for the supplemental information.

THE DATA WILL BE PLACED ON DRYAD IF ACCEPTED

Please apply the use of commas for numbers consistently. For example, compare line 35 and line 38. Also Varroa is not consistently italicized. DONE

I don’t understand why the feral population of bees is mentioned in line 43. You surveyed beekeepers with managed colonies. This has no implications for the feral bees and no evidence is presented that the UK has truly feral colonies. 

DELETED ‘FERAL COLONIES’ AS THE IDEA WAS THAT MORE VARROA-RESISTANT MANAGED COLONIES WILL LEAD TO MORE VARROA-RESISTANT FERAL COLONIES BUT WE AGREE WITH THE COMMENT TO DELETED.

Line 56-57, awkward sentence structure. Unclear what remained high. ENTIRE SECTION REWORDED

Line 60, based used twice REPLACED ONE WITH ESTABLISHED

Line 65-66, not clear why an IPM strategy would be linked to long-term resistance and low mite levels. IPM, as I understand it, is threshold base and involves monitoring and then treating appropriately. How does this encourage low mite levels? And why would this lead to long-term resistance? Needs explanation and evidence. 

THIS IS JUST A REMOTE POSSIBILTY SO THIS STATEMENT HAS BEEN DELETED

Line 102: I have no access to the Figure S1

Line 120: I have no access to Table S1 SORRY THIS HAS OCCURRED

Line 201-203, that assumes that the survey reached and was completed by a representative wedge of the beekeeping population. Many treatment-free beekeepers tend to be younger and more active online, so this may have skewed the responses. These caveats should be mentioned. 

THE COMMENT ABOVE ONLINE ACTIVE HAS NOW BEEN ADDED TO THE RELEVENT PART OF THE DISCUSSION L 239-241. 

Line 218-219: This is not surprising at all. Most treatment free beekeepers in the United States expect annual losses of 50%. Thus, in order to rebuild the following year, they must keep more colonies in reserve. I am disappointed this survey didn’t ask for self-reporting of annual loss rates or honey production per colony. The high level of treatment-free suggests that beekeepers and the much higher colony numbers suggests that the resistance is probably not enough for low annual losses. 

WE DID NOT INCLUDE COLONY DEATHS IN OUR SURVEY SINCE THIS IS ALREADY COVERED BY THE LONG RUNNING BBKA SURVEY DATA FROM WHICH WE HAVE NOW INCULDED(see Fig S2) AND WE HAVE EXPANDED THE PARAGRAPHY COMPAIRING COLONY LOSS RATES FROM UK TREATMENT-FREE BEEKEEPERS. MY PERSONAL EXPERINCE WORKING WORK WITH SEVERAL TREATMENT-FREE BEEKEEPERS THEIR LOSSES ARE SIMILAR TO BEEEKEEPERS THAT ARE TREATING. 

WE DID NOT ASK ABOUT HONEY PRODUCTION AS IN THE UK THIS VARIES GREATLY DUE TO LOCATION AND MICRO-CLIMATE. CLEARLY THERE ARE BIG DIFFERENCES BETWEEN THE SITUATION IN THE UK AND USA IN NUMBER OF TREATMENT FREE BEEKEEPERS THIS IS NOW MENTION IN THE LAST SECTION AS WELL AS THE (Thoms et al )

Line 235-240: This makes the assumption that IPM requires multiple treatments per year. IPM requires monitoring and treating accordingly when a pest level rises above a certain point. Also, many do not consider drone trapping a treatment. As I don’t have an actual copy of the survey questions, I cannot assess if there may have been misunderstandings in the responses. 

POOR PLACEMENT OF SENTENCE SO RE-STRUCTED THIS SECTION AND COMMENTED THAT ONLY VERY BASIC MITE MONTIROING TAKES PLACE.

Line 253-255: This needs more clarification. Is the 4% what beekeepers reported was specifically due to varroa? This would then be an underrepresentation of actual winter losses due to varroa. Numerous surveys have shown that when varroa levels are high, winter losses increase, even though beekeepers blame other factors for the loss. Many beekeepers with colony winter losses don’t know what killed their colony, but beekeepers with more experience can often diagnose the contracted brood nest area and varroa frass. 

WE HAVE GIVEN MORE CLARIFICATION ON THE PRINCIPAL CAUSES OF COLONY LOSSES IN THE UK

Line 254-255: Why is the low winter loss attributed to treatment-free beekeeping. There could be other explanations, such as very active beekeeping groups that teach how to monitor and treat effectively. The evidence presented does not warrant this statement, as there is no segregation of losses by those treating and those not treating presented.

 WE AGREE THAT NOT ENOUGH EVIDENCE IS PRESENT AT THE MOMENT SO THE SENTENCE HAS BEEN DELETED

Line 256-260: These statements require the reader to go to the original articles and assess the numbers of colonies surveyed. Important information such as colonies assessed in the survey should be included with the paraphrased results, especially as these are not peer-reviewed studies but association results and so the statistics have not been evaluated. The way they are presented in this paper, the reader assumes they are peer-reviewed papers, unless the references are checked in detail. 

WE AGREE SO ADDED IN ADDITIONAL DATA/INFO BOTH IN THE TEXT AND SUPP DATA

Line 276-279: This is a very broad statement and is not justified by the results presented, as no losses were calculated in the survey. 

 WE HAVE SHOWEN USING BBKA SURVEY BASED ON 2,613 BEEKEEPERS THAT THERE IS NO SIG RELATIONSHIP BETWEEN %TREATMENT AND %COLONY LOSSES AT A REGIONAL LEVEL IN THE UK. WE HAVE ALSO ADDED IN MORE REFRENCES TO SUPPORT OUR BROAD STATEMENT.

FINALLY THE PAPER HAS BEEN CAREFULLY PROOFED NOW

5. Review Comments to the Author

Reviewer #1: The submitted survey of UK beekeeper´s varroa treatment habits presents interesting new data and fills a gap in our knowledge on recent beekeeping practice. The study is well presented with a clear description of the data sampling and evaluation procedure. The discussion picks up relevant links to related surveys and the establishment of mite resistance in bee populations depending on the treatment strategies. The length of the different chapters is appropriate and well balanced.

Some confusion may arise about the number of respondents for the data in table 3. The table reports on a total of 2,868 responses from 2,897 responses in general (L116). If 25 responses from the Isle of Man have been excluded, some of the 16 participants with zero colonies (L146f) seem to be included here. However, this wouldn´t make any sense for my understanding. The authors should give a precise description on this.

THE 25 RESPONSES FROM THE ISLE OF MAN WERE EXCLUDED FROM THE 2897 (= 2872) WHICH IS NOW STATED IN THE METHODS. IN TABLE 1 THE 16 BEEKEEPERS WITH ZERO COLONIES WHERE EXCLUDED WHICH LEAVES 2856 BEEKEEPERS AS STATED IN TABLE 1. WE HAVE REWRITTEN THE METODS TO INCLUDE ALL SAMPLES EXCULDED 

L102 refers to Fig S1 which is missing in the supplied manuscript. (Sorry our error, CORRECTED)

In L120 Table S1 is mentioned but it should be Table 5 as far as I understand. (deleted)

In L 264 the second “by” has to be erased. Deleted

Reviewer #2: The paper has the potential to be interesting but is currently missing the access to the data and the supplemental information, so difficult to assess. Also, the conclusions are very broad, and the data presented don't support the conclusions the authors make, as the survey did not include any assessment of colony losses. Without distinguishing what the actual loss rates of the treatment free beekeepers have, it is not possible to state that the bees are resistant to varroa or have a high tendency for survival. The higher colony numbers actual suggests the opposite.

AS MENTIONED ABOVE WE HAVE A PARAGRAPH COMPARING LOSSES IN TREATMENT-FREE POPULATIONS VS THE NATIONAL AVERAGE. ANOTHER SURVEY COLLECTS LOSS DATA BUT NOT AIMED AT SEPARATING OUT THE TREATED AND NOT TREATED. WE HAVE ALSO INCLUDED THE MAIN REASONS IN ENGLAND & WALES FOR COLONY LOSSES AND VARROA ACCOUNTS FOR ONLY 4% ALTOUGH WE AGRREE THIS MAY BEEN UNDERESTIMATED BY THE BEEKEEPERS.

Also there are a number of odd sentence structures and grammatical errors. I have made note of the mores specific comments in a separate document. ALL SHOULD BE FIXED NOW

Reviewer #3: The study aims to estimate the number of beekeepers in the UK who are not treating their colonies for varroa, and to estimate whether the existence of untreated and resistant colonies in the UK could be assumed. Information on beekeeping practices is currently limited in the scientific literature, and this question and the data the study provides are relevant and valuable. The article is generally clear and well written. The methodological choices are globally sound and with a high number of respondents to the survey, which is important to note. Still some information should be added in the methodological section, and the limits induced by some methodological choices should be identified and discussed as they can have important impacts on the interpretation of the results. Also, the findings should be discussed regarding the scientific literature on the subject. For these reasons I recommend major revisions.

Major comments

Some methodological limits should be identified and discussed, especially i) the possible influence of the sampling through the beekeeping associations (are these associations involved in prescribing varroa treatments?) 

THERE IS NO PRESCRIPTION OF TREATMENTS VIA ASSOCIATIONS SINCE TREATMENT ADVICE COMES CENTRALLY FROM THE NATIONAL BEE UNIT & THE MAP INDICATES WE HAD A VERY WIDE SPREAD OF RESPONSE 

and ii) the lack of information that the survey provides about the colony losses and about the beginning year of the beekeeping activity. Without this last information, the share of beekeepers who have “never treated” cannot be interpreted as it is done currently regarding the possible existence of surviving colonies (those beekeepers could have started recently, and their colonies may not have faced this absence of treatment for varroa for a long time). It should not be suggested that these colonies had never known any treatment (it could be the case, but the data of the survey do not allow to conclude on this question).

THERE ARE A GROWING GROUP OF BEEEKEEPERS IN THE UK THAT NEVER TREAT SUCH AS OXFORDSHIRE NATURAL BEEKEEPING ASSOCIATION AND THEIR LOSSES FOR TWO WINTERS WERE SIMILAR TO THE NATIONAL AVERAGE. THIS INFORMATION IS PROVIDED IN THE MS AS WELL AS A NEW SUPPLEMENTAL FIGURE SHOWING THE NATIONAL AVERAGE COLONY LOSSES FROM 2007 TO 2021.

The existence of colonies surviving without treatment for varroa and the possible losses that the absence of treatment may induce should also be discussed regarding the scientific literature on these questions.

WE HAVE EXPANDED THE SECTION ON COLONY LOSSESS IN THE UK AS REQUESTED

Other comments

L. 57: “due to almost universal treatment by beekeepers”: this statement should be either nuanced or supported by a reference about American beekeepers’ practices. E.g. see Thoms et al. (2019 - https://doi.org/10.1007/s13280-018-1130-z), which indeed reveals a high percentage of American backyard beekeepers reporting not to treat for varroa.

I CHECKED THOMS ET AL AND ALOUGHT MOST TREATMENT SKEPICS DO NOT TREAT THIS GROUP MADE UP LESS THAN 3% OF THE SURVEY EG (63 VS 2276). THUS, IT APPEARS THAT THERE ARE MUCH FEWER TREATMENT-FREE BEEKEEPERS IN THE USA THAN UK. HOWEVER, WE HAVE CHANGED ‘UNIVERSAL’ TO ‘HIGH’ WHICH IS MORE ACCURATE AND CITED THIS PAPER AT THE END OF THE DISCUSSION. 

L. 100-105: If I understand correctly, there was no question about the colony survival (cf. L100-105). It would have been a valuable information, as the emergence of resistant colonies is only possible if the non-treated colonies survive (in the case of backyard beekeepers replacing regularly their colonies that died from varroa, these non-treated colonies would not be more resistant than other).

SEE EALIER COMMENTS ABOUT COLONY LOSSES

L. 106-113. As the beekeeping associations who forward the survey to their adherents play a major role in the sampling, it would be useful to provide more information about these associations (as such beekeeper associations and their role can differ from one association or one country to the other). Especially, can these associations interfere with their adherents’ treatment choices (e.g., by advising or not advising to treat for varroa)?

THIS IS SOMETHING THAT IS NOT DONE THE ASSOICATIONS ONLY FORWARED THE SURVEY LINK TO ALL ITS MEMBERS USING THERE (CONFEDENTIAL DATABASE). IF THESE TYPE OF BEHAVIOR DID OCCUR, WE WOULD HAVE PICKED IT UP IN THE ANAYLSIS. FURTHERMORE, THE ASSOICATIONS THAT SEND OUT THE LINK HAVE NO WAY OF SEEING WHAT THEIR MEMBERS WORTE. 

L. 123. One of the main analysis choices is the investigation for a possible spatial pattern. As many other factors could possibly influence beekeepers’ choice about their treatment for varroa (beekeeping experience, age, etc), this hypothesis of a spatial pattern could be explained and justified.

EACH ASSOICATION HAS A WIDE RANGE OF AGES, GENDERS, AND EXPERIENCE TO REQUEST THIS TYPE OF DATA AND CONSDIER THEM IN THE ANAYLSIS LIES WAY BEYOND THE SCOPE OF THIS STUDY.

L. 130. Some biomechanical and natural methods that were reported by beekeepers (cf. Table 5) are not efficient against varroa. As the study focuses on the possible existence of untreated / resistant colonies and not on the choice of beekeepers to treat or not to treat (reasons for such choices, etc), these “low-efficacy” treatments could have been considered jointly with the treatment-free group, or a third variant for the response variable could have been considered. So, the choice to gather all the treatments together in a single response variable could be explained.

THE REVIEWER IS CORRECT IN THAT THE DIFFERENT TREATMENTS RANGE WIDLY IN THEIR EFFICACY’S. TO ERR ON THE SIDE OF CAUTION WE HAD TO MAKE IT VERY CLEAR TO UNDERSTAND WHAT ‘TREATMENT-FREE’ MEANT. OTHERWISE, WE COULD HAVE RUN THE PROBLEM OF BEEKEEPERS DECIDING WHAT THEY CONSIDER IS TREATMENT FREE. WE AGREE THE LOW-EFFICACY TREEATMENTS COULD BEE ALLOWING NATURAL RESISTANT TO DEVELOP AND THIS ONE OPTION FOR BEEKEEPERS TO BECOME TREATMENT FREE. IN THE FUTURE IT WOULD BE INTRESTING TO SEE IF THE USAGE OF THESE LOW-EFFICACY METHODS INCREASED. 

L. 133. As one of the questions of the study is the existence of untreated and possibly resistant colonies, it would be useful to know how many colonies in total these 2,872 beekeepers manage and what percentage of the total number of colonies in the UK it represents.

WE HAVE NOW ADDED IN A SECTION AT THE START OF THE RESULTS SHOWING THAT SURVEY ALSO COVERED ABOUT 10% OF THE UK COLONIES 

L. 141-144. This passage is not very clear, especially what is supposed to “explained why there was significant increase in treatment-free beekeepers as the numbers of colonies they managed increased”. It would be useful to rephrase or to complete it.

THIS SECTION HAS BEEEN REEWORDED

L. 146 / Table 1. The third column heading is not very clear: does it represent the percentage of colonies of the group (1-5,…) which are not treated? So, 73% of the colonies of the 16-30 group are not treated?

An additional column providing the percentage of treatment-free beekeepers in each of the groups would be welcome. Besides, the final percentage of untreated colonies (gathering all groups) should be added.

WE RE-CHECKING THE DATA IN TABLE 1 WE DISCOVER AN ERROR SO THANKS FOR THAT. WE HAVE NOW RE-DONE TABLE 1 AND TABLE 2 SO SHOULD BE MUCH CLEARER AS REQUESTED BY THE REVIEWER 

SORRY ITS BEEEKEEPERS NOT COLONIES SO WE HAVE REWORDED IT.

L. 160-162. Grouping the beekeepers who indicated that they have "never treated" with those indicating a specific number of years since the last treatment (here 10 years and more, or 6 years and more L. 201) is questionable as the date of installation of the beekeepers was not in the survey and is not known. Some of them could have started beekeeping recently, and the fact that they have “never treated” for varroa does not presume that their colonies had to face a significant period without treatment. This group of beekeepers with an unknown number of years without treatment should be considered separately and the confusion with the groups where a long treatment-free period is known should be avoided.

IN TABLE 2 WE HAVE SEPRATED OUT THE NEVER ADMINISTRED GROUP SO THEY ARE CONSIDER SEPRATLY. THE MANY OF THESE BEEKEPERS BELONG TO THE OXFORD NATURAL BEEKEEPING. THIS GROUP HAS BEEN AROUND FOR 10 YEARS AND WE PROVIDE COLONY LOSS DATA FROM THIS GROUP AND COMPARE AGAINST THE NATIONAL AVERGAE AND ITS VERY SIMILAR. 

L. 163-164. The fact than some association gather more than 75% of treatment-free beekeepers raises questions about the role of these associations, and about the possible exchanges related to varroa treatments that its members may have. Even if it was not the objective of the study to understand the determinants of the absence of treatment, it would be interesting to discuss the possible role of associations on this point given their central place in the survey sample.

AS PREVIOUSLY MENTIONED VARROA TREATMEENTS ARE RECOMMENDED BY THE GOVERMENTS NATIONAL BEE UNIT. THE TREATMENT-FREE BEEKEEPERS IN MOST ASSOICATIONS DO NOT MENTION THAT THEY DO NOT TREAT AS THEY ARE IN MINORITY HENCE WHY WE DID THE SURVEY TO GET A BETTER ESITATMTION OF THEE THE NUMBER OF TREATMENT FREE BEEKEEPERS.

L. 178. “the majority are treated”: the majority of colonies? YES, SO CHANGED

L. 180-184. It would be useful to add the global percentage of beekeepers not using any treatment and to more clearly distinguish if the percentages given for the types of treatments are exclusive of each other or not. E.g., can the 3% of beekeepers using biomechanical methods be also in the 78% using chemical? Or are they only using biomechanical methods?

THIS DATA DOES NOT CURRENTLY EXIST THE Thoms et al survey and this study are some of the first attempts to collect such data.

L. 201. See comment on L. 160-162

L. 202. The extrapolation of the study results to the 30,000 estimated UK beekeepers should be supported by a discussion about the representativity of the sample. That joins the question of giving information about the beekeeping associations (see comment on L 103-115), and about their representativeness (are they some apicultural practices which may differ between members of these associations and beekeepers who are not members, e.g., if the associations are involved in prescribing treatments for varroa? Are the members of these associations representing a large share of the UK beekeepers?)

IN THE UK THERE ARE AROUND 30,000 AMATEUR BEEKEEPERS BUT ONLY 400 PROFESSIONALS THAT BELONG TO THE BEE FAMERS SOCIETY. THIS IS NOW MENTIONED IN THE METHODS.

L. 257-260. As varroa is usually considered in scientific and technical literature as an important factor of colony losses, some scientific references about the place of varroa in colony losses need to be added and the technical references which are the only provided references should be discussed regarding these scientific references.

THIS IS VERY TRUE PARTICULAR IN THE NORTHERN HEMISPHERE BUT IN MANY COUNTRIES (E.G., AFRICA, BRAZIL-MEXICO, CUBA) VARROA IS NOW NOT CONSIDERED A PROBLEM SINCE THE BEES ARE ABLE TO KEEP THE INFESTATION LEVEL DOWN BELOW 10-4% DEPENDING ON THE POPULATION. HENCE IN THE EYES OF WELL-ESTABLISHED (6YRS+) TREATMENT-FREE BEEKEEPERS VARROA IS NOT CONSIDERED AS A SERIOUS PROBLEM. THE SAME CAN BE SAID BY BEEKEEPERS THAT REGULAR TREAT THEIR VARROA POPULATIONS. WE HAVE NOW ADDED IN A SHORT SECTION SHOWING IN ONE SURVEY THE BEEKEEPERS REPORTED THAT ONLY 4% OF THEIR DEATHS WAS DIRECTLY CAUSED BY VARROA.

L. 267. It would be useful to precise “in Hawaii” and not only “in the USA” as the context of an island can be specific. CHANGED AS SUGGESTED

Fi. 1. A. A legend with the number of respondents corresponding to the different pie sizes is needed here. THIS HAS NOW BEEN DONE

---

## [Decision Letter · Decision Letter 1]

7 Dec 2022

PONE-D-22-05221R1A survey of UK Beekeeper’s Varroa treatment habitsPLOS ONE

Dear Dr. Martin,

Thank you for submitting your manuscript to PLOS ONE. After careful consideration, we feel that it has merit but does not fully meet PLOS ONE’s publication criteria as it currently stands. Therefore, we invite you to submit a revised version of the manuscript that addresses the points raised during the review process.

We look forward to receiving your revised manuscript.

Kind regards,

Vanessa Corby-Harris, Ph.D.

Academic Editor

PLOS ONE

Additional Editor Comments:

Dear authors,

(First, I apologize for my lengthy response. This manuscript was re-assigned to me and so I was reading it for the first time.)

This manuscript summarizes a survey of whether UK beekeepers treat for mites or not. Overall, I think that this manuscript could be a good fit for PLoS One based on the journal’s publication criteria, with one exception noted below. I also think that the general idea of treatment-free beekeeping is interesting and that it would be a useful addition to the literature. However, I do agree with reviewer 2 that the manuscript is somewhat narrow in scope and rather descriptive, and would be better suited for a more specialized or beekeeper-friendly journal. However, many of those journals are pay walled and so this could be a better way to more widely disseminate the authors’ results.

This manuscript has been through one round of review and was much improved after the reviewer comments were addressed. You will see that the reviewers have some lingering concerns that still need to be addressed. Please address each comment in your revision. Specifically, reviewers 2 and 3 have multiple concerns, such as the rationale behind several hypotheses, statements that need supporting citations, the level of detail in the methods, and concerns about the survey itself. There are also several instances where the conclusions are not supported by the data (ex. treatment-free beekeeping is becoming more common). Please revise those statements so that the manuscript meets the PLoS One publication criteria, specifically criteria #4 that “Conclusions are presented in an appropriate fashion and are supported by the data”.

I also saw several small grammatical errors and had a few minor questions. I apologize that these are coming up now, but they need to be addressed.

1) There are several sentences that start with “Although”, “Thus”, “Whereas”. Please revise these sentences because they are often grammatically incorrect.

L56 – decades

L59 – I wouldn’t say they are highly effective. For example, most or all can’t get into the brood cell, where mite reproduction happens. Revise?

 L61-62 – watch what you capitalize (ex. oxalic?)

L65 – take out the comma after reference 9

L83 – is this resistance or tolerance?

L103 – six questions:

L134 – isn’t “treatment status” the response variable?

L140 – widely or wildly?

L141-143 – this should be in the methods and needs more detail (see reviewer 2 comment)

Figure 1 – it could be helpful to remind the reader that the 158 area associations in the spatial study had ≥5 respondents, so associations with <5 respondents were not colored in black. At least that was my interpretation of what you said in the text.

L160-161 – Is this an important or real trend if you don’t see a difference between these groups and the larger groups? Can you explain or speculate why you didn’t see differences between these small and medium colony number groups and the larger group?

Table 2 – had no effect (see legend text); also should you have a range for the colony number group estimate since you had four groups?

I like Tables 4 and 5, although I agree with reviewer 3 that there are issues with the “treatment-free” group. Some of those natural methods were interesting (rhubarb??). And how does queen trapping differ from a brood break? Just wondering, really.

L218-219 – This is repeating part of my point above, but I agree that this group could also include beekeepers that have kept bees for only a few (<6) years and so combining them with the beekeepers that have been treatment-free for 6 years or more could be misleading. It seems that there was no criteria that respondents have kept bees for 6 or more years.

L275-276 – revise the sentence where it says weather-related and Varroa – should this be split up somehow?

L281- that’s not a lot of data (N=6) to base a conclusion on. I am also of the opinion that the discussion should not include too many results/statistics, so this is something to potentially omit. It is appropriate to talk about it, just not as something that you tested using data and stats.

L307-310 – revise this sentence, it is too long. I like the general idea of the sentence though!

Reviewers' comments:

Reviewer's Responses to Questions

**Comments to the Author**

1. If the authors have adequately addressed your comments raised in a previous round of review and you feel that this manuscript is now acceptable for publication, you may indicate that here to bypass the “Comments to the Author” section, enter your conflict of interest statement in the “Confidential to Editor” section, and submit your "Accept" recommendation.

Reviewer #1: All comments have been addressed

Reviewer #2: All comments have been addressed

Reviewer #3: (No Response)

2. Is the manuscript technically sound, and do the data support the conclusions?

Reviewer #1: Yes

Reviewer #2: Yes

Reviewer #3: Partly

3. Has the statistical analysis been performed appropriately and rigorously? 

Reviewer #1: Yes

Reviewer #2: Yes

Reviewer #3: Yes

4. Have the authors made all data underlying the findings in their manuscript fully available?

Reviewer #1: Yes

Reviewer #2: Yes

Reviewer #3: No

5. Is the manuscript presented in an intelligible fashion and written in standard English?

Reviewer #1: Yes

Reviewer #2: Yes

Reviewer #3: Yes

6. Review Comments to the Author

Reviewer #1: please correct the following typos in the revised document:

L168: ...had no effect...

L270: overwinter losses. Over the last...

L307: ...than found currently in the UK.

Regarding your statement in L249-251I suggest a more careful wording as the BBKA report is restricted on the treatment period between August and September while OA as the most common treatment in UK should be mainly used on broodless colonies during winter.

Reviewer #2: The paper is well written, but the survey lacks depth. The authors wished to investigate how wide spread treatment free beekeeping is within the UK. They did this by conducting an online survey with limited questions that could be completed within five minutes. It would have been far more interesting if they then did a follow up with the beekeepers who qualified as treatment-free to learn more about their beekeeping practices. I feel the current results presented are more appropriate for a bee journal than a rigorous scientific publication. For specific comments regarding the manuscript, see attached PDF.

Reviewer #3: SPECIFIC COMMENTS

L. 72 References are still missing to support this assertion

L 127-132 The choice to explore a potential spatial pattern (rather than the other possible explanatory factors for the varroa management) should still be explained: why is this hypothesis made?

L 233. References or studies on the number of treatment free beekeepers a few years ago would be appropriate here to support the assertion that the number of treatment-free beekeepers is growing.

L 240-244 The BBKA studies the authors refer to are about a specific time of the year: even if some of the beekeepers who do not treat at this time of the year could be treatment-free beekeepers as the authors underline, some of them could also treat at another time of the year. Thus, the results of these BBKA studies should not be directly used to compare with the percentage of treatment-free beekeepers which was estimated in the study, as they do not estimate the same thing.

L.279 It is unclear if the percentage of colonies not treated here refers to the “august to September non treated” (cf. L 241) or to year-long non treated (and the BBKA study the authors refer to does not seem to be available online to find this information). It would be appropriate to precise this information.

GENERAL COMMENTS

This study provides useful information about the number of treatment-free beekeepers in the UK. Some relevant changes have been made to the paper, considering the reviewers’ feedback on the first manuscript.

Still, the authors did not respond to some of the feedback, in particular regarding their methodological choices, which were not always explained (e.g. why explore a spatial pattern?) and regarding the limits of these methodological choices and therefore of the study.

One of the major methodological issues is the choice to gather beekeepers who indicated that they have “never treated against varroa” with beekeepers who have not treated for six years or more. This choice relies on the hypothesis that beekeepers who have “never treated” manage colonies which have not been treated for a significant period of time (6 years or more, or a least several years). This cannot be stated, as the survey lacks a question on the duration of the colony management by beekeepers: the beekeepers who have “never treated” could also be beekeepers who just started beekeeping in the year the survey was made. The number of years in which their colonies would have been treatment-free is then totally unknown, and could also be less than a year.

Considering this issue, the number of treatment-free beekeepers have to be estimated without this “never treated” category.

Also, one of the main conclusions of the authors seems to be that treatment-free beekeeping would be increasing and that it would be a reliable solution as the colony losses would not be greater: these conclusions are not supported by the study as the survey was not designed properly to study these questions. It would have required questions about the colony losses and about the age of the beekeeping activity. These questions were not asked in the survey and the references that the authors provide to support these conclusions are clearly not sufficient to conclude about this. Indeed, the compared category is not the same for the BBKA study, and the given references e.g. the Oxfordshire Beekeeping Group are particular cases and not scientific literature. This does not seem to be strong enough arguments to support such an important conclusion as the supposed absence of additional colony losses in the case of treatment-free beekeeping.

There are of course relevant questions and perspectives that this study opens up. Still, it should be seen as such and not considered as actual and current results.

To conclude on that new version of the manuscript, some efforts were made by the authors since the first draft and these efforts must be acknowledged. Still, it seems that this manuscript still requires some modifications to address properly the methodological issues and to ensure a better and clear distinction between what is an actual conclusion of the study and what remains a perspective without sufficient evidence to conclude here.

7. PLOS authors have the option to publish the peer review history of their article (what does this mean?). If published, this will include your full peer review and any attached files.

Reviewer #1: No

Reviewer #2: No

Reviewer #3: No

---

## [Author Response · Author response to Decision Letter 1]

20 Dec 2022

1) There are several sentences that start with “Although”, “Thus”, “Whereas”. Please revise these sentences because they are often grammatically incorrect.

L56 – decades CORRECTED

L59 – I wouldn’t say they are highly effective. For example, most or all can’t get into the brood cell, where mite reproduction happens. Revise? REVISED

 L61-62 – watch what you capitalize (ex. oxalic?) DONE

L65 – take out the comma after reference 9 DONE

L83 – is this resistance or tolerance? SEVERAL YEARS A GO I USED TOLERANCE BUT WAS TOLD TO USE RESISTANCE WHEN TALKING ABOUT THIS MECHANSIM. IN FACT BOTH TERMS CAN APPLY DEPENDING WHAT VIEW YOU TAKE. THEREFORE, I NOW USE RESISTANCE AS THIS IS THE DOMINANT TERM USED IN THE FIELD AND DEFINE IT AS I HAVE DONE HERE.

L103 – six questions: DONE

L134 – isn’t “treatment status” the response variable? YOU ARE CORRECT SO CHANGED

L140 – widely or wildly? WIDELY IS THE CORRECT TERM 

L141-143 – this should be in the methods and needs more detail (see reviewer 2 comment) DONE AND ADDED NEW DETAIL REQUESTED

Figure 1 – it could be helpful to remind the reader that the 158 area associations in the spatial study had ≥5 respondents, so associations with <5 respondents were not colored in black. At least that was my interpretation of what you said in the text. WE HAVE NOW ADDED THIS INFO INTO THE LEGEND

L160-161 – Is this an important or real trend if you don’t see a difference between these groups and the larger groups? Can you explain or speculate why you didn’t see differences between these small and medium colony number groups and the larger group?

FURTHER ANAYLSIS (by calculating the 95% CI) INDICATED THAT DUE TO A HIGH VARIABILITY IN THE DATA IN THE DATA WE CAN CONCLUDE THERE IS NO SIGNIFICANT RELATIONSHIP BETWEEN COLONY NUMBER AND TREATMENT STATUS. THIS IS NOW MENTIONED.

Table 2 – had no effect (see legend text); also should you have a range for the colony number group estimate since you had four groups? ALEX

I like Tables 4 and 5, although I agree with reviewer 3 that there are issues with the “treatment-free” group. Some of those natural methods were interesting (rhubarb??). And how does queen trapping differ from a brood break? Just wondering, really.

TABLE 4 WE HAVE CHANGED TO ‘NOT TREATING’ AS THIS IS WHAT THEY WERE DOING AT THE TIME OF THE SURVEY, BUT WE DO AGREE WITH YOU AND REVIEWER 3 THAT ELSEWHERE WE BREAK THIS GROUP UP INTO ITS VARIOUS GROUPS AS WE HAVE DONE IN TABLE 3. 

AS FOR TABLE 5 WE COULD COMBINE BROOD BREAKS WITH QUEEN TRAPPING BUT THESE ARE THE TERMS THE BEEKEEPERS USED IN THE SURVEY.

RHUBARD LEAVES CONTAIN OXALIC ACID BUT THE LEVEL WILL BE LOW

L218-219 – This is repeating part of my point above, but I agree that this group could also include beekeepers that have kept bees for only a few (<6) years and so combining them with the beekeepers that have been treatment-free for 6 years or more could be misleading. It seems that there was no criteria that respondents have kept bees for 6 or more years.

IN HINDSIGHT THIS WAS A OVERSIGHT OF THE QUESTIONS ASKED AND RESPONSE OPTIONS AVAILABLE. BUT NOW WE HAVE SEPARATED OUT THE GROUP OF NEVER TREATED AND 1-5 YEARS GROUP. THE READER SHOULD NOW BE ABLE TO DISTINGUSH BETWEEN THOSE NOT-TREATING AND THOSE THAT ARE TREATMENT FREE (I.E. 6+ YEARS) 

L275-276 – revise the sentence where it says weather-related and Varroa – should this be split up somehow?

DONE

L281- that’s not a lot of data (N=6) to base a conclusion on. I am also of the opinion that the discussion should not include too many results/statistics, so this is something to potentially omit. It is appropriate to talk about it, just not as something that you tested using data and stats.

SECTION REMOVED

L307-310 – revise this sentence, it is too long. I like the general idea of the sentence though!

SPLIT INTO 2 SENTENCES 

Reviewer #1: please correct the following typos in the revised document:

L168: ...had no effect... CORRECTED

L270: overwinter losses. Over the last...CORRECTED

L307: ...than found currently in the UK…CORRECTLY

Regarding your statement in L249-251I suggest a more careful wording as the BBKA report is restricted on the treatment period between August and September while OA as the most common treatment in UK should be mainly used on broodless colonies during winter.

THE BBKA SURVEYS ALSO COVERS THE OCT TO APRIL PERIOD AS WELL SO THIS DATA IS NOW GIVEN IN ADDITION TO THE AUG-SEPT DATA.

Reviewer #2: The paper is well written, but the survey lacks depth. The authors wished to investigate how wide spread treatment free beekeeping is within the UK. They did this by conducting an online survey with limited questions that could be completed within five minutes. It would have been far more interesting if they then did a follow up with the beekeepers who qualified as treatment-free to learn more about their beekeeping practices. I feel the current results presented are more appropriate for a bee journal than a rigorous scientific publication. For specific comments regarding the manuscript, see attached PDF.

ALL COMMENTS ADRESSED ON ATTACHED PDF (see attached PDF)

Reviewer #3: SPECIFIC COMMENTS

L. 72 References are still missing to support this assertion

IT IS NOW MADE CLEAR THAT THERE IS NO EVIDENCE TO SUPPORT OR CONTRADICT THIS IDEA. 

L 127-132 The choice to explore a potential spatial pattern (rather than the other possible explanatory factors for the varroa management) should still be explained: why is this hypothesis made?

WE HAVE INSERTED THE FOLLOWING TEXT ‘WHILE CONDUCTING PREVIOUS RESEARCH INTO VARROA-RESISTANCE COLONIES [14] IT APPEAR THEY EXISTED IN MANY PARTS OF THE UK. THEREFORE, MAPS WERE CREATED……’ 

L 233. References or studies on the number of treatment free beekeepers a few years ago would be appropriate here to support the assertion that the number of treatment-free beekeepers is growing.

THIS SECTION HAS BEEN DELETED AS THIS STUDY IS MEANT TO PROVIDE BASE-LINE DATA FROM WHICH WE CAN SEE IF IN THE FUTURE THINGS INCREASE OR DECREASE. 

L 240-244 The BBKA studies the authors refer to are about a specific time of the year: even if some of the beekeepers who do not treat at this time of the year could be treatment-free beekeepers as the authors underline, some of them could also treat at another time of the year. Thus, the results of these BBKA studies should not be directly used to compare with the percentage of treatment-free beekeepers which was estimated in the study, as they do not estimate the same thing.

WE HAVE INCLUDED ADDITIONAL ‘NON-TREATMENT’ DATA (OCT-APRIL) FROM THE BBKA SURVEY 

L.279 It is unclear if the percentage of colonies not treated here refers to the “august to September non treated” (cf. L 241) or to year-long non treated (and the BBKA study the authors refer to does not seem to be available online to find this information). It would be appropriate to precise this information.

TO ACCESS THE DATA YOU MUST FIRST JOIN THE BBKA IN ORDER TO ACCESS THERE NEWSLETTER INCLUDING THE BRITISH BEE JOURNAL, SINCE THE JOURNAL/NEWSLETTER LIKE MANY JOURNALS IS BEHIND A PAYWALL.

GENERAL COMMENTS

This study provides useful information about the number of treatment-free beekeepers in the UK. Some relevant changes have been made to the paper, considering the reviewers’ feedback on the first manuscript.

Still, the authors did not respond to some of the feedback, in particular regarding their methodological choices, which were not always explained (e.g. why explore a spatial pattern?) and regarding the limits of these methodological choices and therefore of the study.

THE REASON HAS NOW BEEN GIVEN

One of the major methodological issues is the choice to gather beekeepers who indicated that they have “never treated against varroa” with beekeepers who have not treated for six years or more. This choice relies on the hypothesis that beekeepers who have “never treated” manage colonies which have not been treated for a significant period of time (6 years or more, or a least several years). This cannot be stated, as the survey lacks a question on the duration of the colony management by beekeepers: the beekeepers who have “never treated” could also be beekeepers who just started beekeeping in the year the survey was made. The number of years in which their colonies would have been treatment-free is then totally unknown, and could also be less than a year.

THE ‘NEVER TREATED’ GROUP IS NOW TREATED SEPARATLY 

Considering this issue, the number of treatment-free beekeepers have to be estimated without this “never treated” category. 

THIS HAS NOW BEEN DONE

Also, one of the main conclusions of the authors seems to be that treatment-free beekeeping would be increasing and that it would be a reliable solution as the colony losses would not be greater: these conclusions are not supported by the study as the survey was not designed properly to study these questions. It would have required questions about the colony losses and about the age of the beekeeping activity. These questions were not asked in the survey and the references that the authors provide to support these conclusions are clearly not sufficient to conclude about this. Indeed, the compared category is not the same for the BBKA study, and the given references e.g. the Oxfordshire Beekeeping Group are particular cases and not scientific literature. This does not seem to be strong enough arguments to support such an important conclusion as the supposed absence of additional colony losses in the case of treatment-free beekeeping.

There are of course relevant questions and perspectives that this study opens up. Still, it should be seen as such and not considered as actual and current results.

WE HAVE REDUCED AND RE-WRITTEN THAT SECTION AND ADDED THAT AN ‘independent scientific studies are required to see if Varroa-resistance effects the rate of colony losses.’

---

## [Editor Report · Decision Letter 2]

16 Jan 2023

A survey of UK Beekeeper’s Varroa treatment habits

PONE-D-22-05221R2

Dear Dr. Martin,

We’re pleased to inform you that your manuscript has been judged scientifically suitable for publication and will be formally accepted for publication once it meets all outstanding technical requirements.

Kind regards,

Vanessa Corby-Harris, Ph.D.

Academic Editor

PLOS ONE

Additional Editor Comments (optional):

After carefully reading through the reviewers' comments on the first revision and the authors' response to these comments, I feel this article is acceptable.

---

## [Editor Report · Acceptance letter]

20 Jan 2023

PONE-D-22-05221R2 

A survey of UK Beekeeper’s *Varroa* treatment habits 

Dear Dr. Martin:

I'm pleased to inform you that your manuscript has been deemed suitable for publication in PLOS ONE. Congratulations! Your manuscript is now with our production department. 

Kind regards, 

on behalf of

Dr. Vanessa Corby-Harris 

Academic Editor

PLOS ONE